# On-Demand Metallization System Using Micro-Plasma Bubbles

**DOI:** 10.3390/mi13081312

**Published:** 2022-08-13

**Authors:** Yu Yamashita, Shinya Sakuma, Yoko Yamanishi

**Affiliations:** Department of Mechanical Engineering, Kyushu University, Fukuoka 819-0395, Japan

**Keywords:** collapse of bubble, plasma discharge, metal deposition, reduction, wiring

## Abstract

3D wiring technology is required for the integration of micro–nano devices on various 3D surfaces. However, current wiring technologies cannot be adapted to a variety of materials and surfaces. Here, we propose a new metal deposition method using only a micro-plasma bubble injector and a metal ion solution. Micro-plasma bubbles were generated on demand using pulses, and the localized reaction field enables metal deposition independent of the substrate. Three different modes of micro-plasma bubble generation were created depending on the power supply conditions and mode suitable for metal deposition. Furthermore, using a mode in which one bubble was generated for all pulses among the three modes, copper deposition on dry/wet materials, such as chicken tissue and glass substrates, was achieved. In addition, metal deposition of copper, nickel, chromium, cobalt, and zinc was achieved by simply changing the metal ion solution. Finally, patterning on glass and epoxy resin was performed. Notably, the proposed metal deposition method is conductivity independent. The proposed method is a starting point for 3D wiring of wet materials, which is difficult with existing technologies. Our complete system makes it possible to directly attach sensors and actuators to living organisms and robots, for example, and contribute to soft robotics and biomimetics.

## 1. Introduction

Advances in 3D fabrication technologies, such as 3D printing, have led to many structural functions that take advantage of their shape and mechanical properties [1,2,3]. Recently, micro/nano-devices with extended functions with incorporated and integrated sensors [4,5] and actuators [6,7] were reported with excellent features, such as a small size, high sensitivity, and high resolution. These devices are typically fabricated by conventional 2D fabrication technologies, such as micro-electromechanical systems’ fabrication techniques. To integrate micro/nano-devices on 3D microstructures, 3D wiring technology is required for wiring, connections, signal transmission, and power supply. Electro/electroless plating, PVD, and CVD are widely used as conventional wiring technologies; however, each process has limitations in terms of material conductivity and heat resistance. In addition, patterning requires a complicated process, which is not suitable for local wiring on a 3D surface [8]. Inkjet printing combined with automatic scanning printing may be one of the most promising methods for patterning on 3D surfaces because it does not require a mask, has a high resolution, and high speed [9,10]. However, it is necessary to consider oxidation of the wiring and the heat resistance of the substrate because sintering is used after wiring. Therefore, for localized wiring on any 3D surface, an innovative method that does not depend on dry/wet environments or conductive/non-conductive properties is needed. One promising method to solve these problems is to use a plasma reaction field. Radiation [11,12], lasers [13,14], radiofrequency pulses [15], microwaves [16], and atmospheric pressure gas plasmas [17,18] are reported as plasma sources for producing metallic nanoparticles. These plasmas can form localized reaction fields. In particular, plasmas generated in solution have the potential for high-speed metal deposition while avoiding thermal damage by liquid cooling and an abundant ion source. The strategy required is to control the plasma locally and on-demand.

Here, we propose a new metal deposition method using micro-plasma bubbles generated by an applied pulse voltage. In this study, we first classify the generation modes of micro-plasma bubbles according to the conditions of the applied pulse. Next, the effect of the generation mode on the probability and resolution of the metal deposition is verified with an in situ observation system. Finally, we demonstrate metal deposition using this method and confirm the feasibility of on-demand wiring on non-conductive materials.

## 2. Concept of Electrical Patterning

Figure 1 shows the proposed metal deposition system using micro-plasma bubbles. The system comprises a bubble injector with a dielectric capillary and a wire electrode as the active electrode, a counter electrode, a metal ion solution, and a pulsed power supply. When a pulse voltage is applied to the bubble injector, a high electric field is formed by concentrating the electric field at the tip of the bubble injector, as shown in Figure 1a. The high electric field quickly breaks down gases, such as dissolved gases and water vapor, by local heating and electrochemical gases [19]. Therefore, we can generate micro-plasma bubbles via a rapid increase in temperature and pressure at the micro-region because of the high electric field [20]. When micro-plasma bubbles are generated in an aqueous solution, they produce active species, such as hydrated electrons and hydrogen atoms, and ionic species such as oxonium ions. These active species and ion species make it possible for a specific local reaction field to form. Specifically, hydrated electrons and hydrogen atoms have a powerful reducing effect and can reduce metal ions in solution, as shown in Equations (1) and (2) [21,22,23]. Furthermore, the polarity of the bubble injector also has an effect. Bubble injectors are usually connected to either the anode or the cathode. Electrochemical reaction via plasma which induces a reduction on the cathode side and oxidation on the anode side. Thus, gold nanoparticles and magnetite nanoparticles can be formed [24,25]. Our method can be realized without the use of helium or other gases during plasma generation. In addition, micro-plasma bubbles expand and then collapse to produce a localized flow called a microjet, as shown in Figure 1c [26]. The reduced metal agglomerates on the surface because of the micro-plasma bubble expansion/collapse pressure, as shown in Equation (3). We form deposition patterns comprising arbitrarily shaped and metallic species by continuously scanning the substrate using the physicochemical reactions of the micro-plasma bubbles. The proposed method may be applied to various materials, such as dry/wet and conductive/non-conductive materials, because it only involves electrical writing with a bubble injector and metal ion solution.
(1)$${\mathrm{ne}}_{\mathrm{aq}}^{-}+M^{n+}\to {\mathrm{nH}}_{2}O+{\mathrm{ne}}^{-}+M^{n+}\to {\mathrm{nH}}_{2}O+M^{0}$$
(2)$${\mathrm{nH}}^{∙}+M^{n+}\to {\mathrm{nH}}^{+}+{\mathrm{ne}}^{-}+M^{n+}\to {\mathrm{nH}}_{2}O+M^{0}$$
(3)$${\mathrm{nM}}^{0}\to \frac{n}{2}M_{2}\to \cdots M_{n}\ldots \to {\mathrm{nM}}_{\mathrm{agg}}$$

In the above equations, e_aq_^−^, H, M^n+^, M and M_agg_ are hydrated electrons, hydrogen atoms, metal ions, metal atoms, and metal aggregates, respectively.

## 3. Materials and Methods

### 3.1. Experimental Setup

Figure 2 shows a schematic and photograph of the experimental setup. Our apparatus comprises a bubble injector, counter electrode, pulse power supply (PLT1500, BEX Co., Ltd., Tokyo, Japan), 1 kΩ resistor, 6-axis manipulator, oscilloscope (DS5614A, Iwatsu Electric Co., Ltd., Tokyo, Japan), high-speed camera (HPV-X2, Shimadzu Corporation, Kyoto, Japan), microscope (VHX, Keyence Corporation, Osaka, Japan), and light source. Three experiments were conducted: classification of the micro-plasma bubble generation modes by pulse conditions, investigation of the effects of the generation mode on the probability and resolution of metal deposition, and a wiring demonstration. Two types of bubble injectors were used. Type A used a glass capillary with an inner diameter of 130 μm and an outer diameter of 500 μm, into which a 100 μm tungsten wire was inserted, and the capillary was fixed so that the end faces of the tip were aligned. Type B used a glass capillary with an inner diameter of 500 μm and an outer diameter of 1000 μm, into which a 400 μm tungsten wire was inserted and the capillary tip was constricted to form a void with a diameter of 300 μm and a height of 300 μm. Type A devices were used in the bubble generation classification and metal deposition evaluation experiments, while both devices were used in the metal deposition demonstration. For the metal ion solutions, a copper sulfate solution adjusted to 1 mol/L was used in the bubble generation classification and metal deposition evaluation experiments. For the metal deposition demonstration, copper acetate solution, nickel acetate solution, cobalt acetate solution, and zinc acetate solution adjusted to 0.3 mol/L were used in addition to a 1 mol/L copper sulfate solution. A chromium acetate solution adjusted to 50 g/L was used. A 6-axis manipulator precisely controlled the position and angle of the bubble injector. The jig for grasping the bubble injector and the chamber for setting the substrate was fabricated by a 3D printer. The bubble injector was connected to the cathode and the counter electrode to the anode.

### 3.2. Classification of Bubble Generation

First, we classified the generation mode of micro-plasma bubbles by analyzing the number of pulses applied to the micro-plasma bubbles from generation to collapse. We observed the micro-plasma bubbles using a high-speed camera and compared the behavior of micro-plasma bubbles with pulse waveforms obtained using an oscilloscope. The behavior of micro-plasma bubbles was evaluated using time series data of the bubble diameter, which was calculated by binarizing the difference between the first frame and each frame of the movie and considering the projected area as a circle. In the experiments, the off time varied from 2.5 µs to 100 µs, the voltage ranged from 600 V to 1000 V in 100 V steps, and the pulse widths were 5 µs, 10 µs and 15 µs. The number of pulses was set to 10 for all conditions.

### 3.3. Evaluation of the Metal Deposition

Next, we varied the generation mode of the micro-plasma bubbles by changing the off time and evaluated the probability and resolution of metal deposition under each condition. We constructed an in situ observation system using a microscope to observe the metal deposition. The voltage was set to 700 V, the pulse widths to 5 µs, 10 µs, and 15 µs, and the number of pulses to 2000, 1000, and 667, respectively, so that the product of the pulse width and the number was approximately 10 ms. The deposition probability was evaluated by dividing the number of samples in which the number of trials confirmed deposition. The resolution of the deposits was evaluated by the diameter of the deposits, which was calculated by considering the area of the deposits as a circle based on the binarization process. We used a glass substrate with excellent transparency (Matsunami micro cover glass, Matsunami Glass Ind., Ltd., Osaka, Japan). The distance between the device and substrate was set to 200 µm. The reproducibility was confirmed by performing each condition five times.

### 3.4. Demonstration of Electrical Patterning

Finally, we performed copper deposition on a variety of substrates and various types of metal deposition on epoxy resin to demonstrate metal depositions. The deposited metals were analyzed using energy dispersive X-ray spectroscopy. In addition, metal deposition and bubble injector movement were alternated to write wiring on the glass substrates. The experiment was performed using the same in situ apparatus as in Section 3.3. The voltage, pulse width, and off time were set to the conditions that resulted in the APSB mode.

## 4. Results

### 4.1. Classification of the Bubble Generation

The modes of micro-plasma bubble generation were classified into three categories: single pulse to single bubble (SPSB); multiple pulses to single bubble (MPSB); and all pulses to single bubble (APSB). Figure 3a–c show typical examples of SPSB, MPSB, and APSB, with a voltage of 700 V, a pulse width of 10 µs, and off times of 100 µs, 50 µs, and 10 µs, respectively. In the SPSB mode (Figure 3a), a single bubble is generated at the tip of the bubble injector with the first pulse. The bubble diameter reached approximately 700 µm at 50 µs after the pulse was applied, and then the bubble collapsed before the next pulse was applied. In the MPSB mode (Figure 3b), the first bubble was generated by the first pulse, and the second pulse was applied before the first bubble collapsed, which caused the first bubble to expand. The diameter of the first bubble reached approximately 700 µm approximately 50 µs after the pulse was applied, and it expanded slightly when the next pulse was applied at 60 µs. Then, at approximately 100 µs, the bubble collapsed. In the APSB mode (Figure 3c), a single bubble was generated with the first pulse. The bubble diameter gradually increased as the pulse was applied, reaching a maximum diameter of approximately 1140 µm at 220 µs. When the pulse was stopped, the micro-plasma bubble began to shrink and finally collapsed at 310 µs. Thus, in the APSB mode, the micro-plasma bubble was held at the tip of the bubble injector while the pulse was applied, and it collapsed when the pulse was stopped.

Figure 4 shows the mapping results of the bubble generation modes for each voltage, pulse width, and off time. The boundaries of APSB, MPSB, and SPSB are indicated by dashed lines. The boundary between APSB and MPSB is between 10 µs and 40 µs off times at each pulse width. The larger the voltage, the longer the pulse width and the larger the boundary shifts to longer off times. The boundary between MPSB and SPSB exists over a wide range of off times, from 30 µs to more than 100 µs: the larger the voltage, the longer the pulse width, shifting the boundary to longer off times. Therefore, the off time is a critical parameter that determines the mode of micro-plasma bubble generation because it is essential to know when the next pulse is applied during the bubble generation/expansion/collapse.

### 4.2. Evaluation of the Metal Deposition

It was possible to deposit metallic copper in the range of 100 µm to 400 µm on glass substrates by using the APSB mode. Figure 5 shows the results of the deposition probability and deposit diameter for each pulse width and off time, including a photograph of a typical deposit. The mode of micro-plasma bubble generation is also shown in this figure. High probability deposition was confirmed for pulse widths of 5 µs and 10 µs when the off time was less than 20 µs, and for pulse widths of 15 µs when the off time was less than 30 µs. For all pulse widths, deposition became unstable as the off time became longer, and no deposition pattern could be observed above 40 µs. The deposit diameter was approximately 200 µm for all pulse widths. These results show that the off time is an essential parameter for writing metal patterns. Additionally, the pulse width and off time do not significantly affect the deposit diameter and thus, the wiring resolution. Because the generation mode of micro-plasma bubbles for APSB is less than 15 µs and MPSB is longer than 15 µs, it is assumed that there is a close relationship between metal deposition and the generation mode of a micro-plasma bubble. In the bubble observation experiment, micro-plasma bubbles were generated without a substrate in front of the bubble injector. However, in this experiment, the substrate may have made it easier for bubbles to accumulate at the tip of the bubble injector. Therefore, in the off-time range of 20 µs to 30 µs, the bubbles may have been generated in the APSB mode instead of the MPSB mode. These results confirm that Cu was successfully deposited on a glass substrate, which is a non-conductive material, by setting the off time to the APSB mode.

### 4.3. Demonstration of Electrical Patterning

We achieved metal deposition on various substrates and the deposition of various metal species by generating micro-plasma bubbles in the APSB mode. Figure 6 shows representative microscopic images of metal depositions on various substrates. Using the proposed method, we confirmed metal deposition on dry/wet materials, such as chicken tissue, acrylic resin, silicon wafers, nitrile rubber, and glass. Figure 7 show the deposition of various metal species and their elemental analysis. Figure 7c also shows the percentage of each element. The deposition of copper, nickel, chromium, cobalt, and zinc metals was possible using each metal ion solution. However, as the electron microscope images clearly show, the film was not uniform, with cracks and unevenness. On the other hand, as shown in the results obtained by subtracting the substrate fraction from each elemental fraction, all but chromium exhibited low oxidation rates. In particular, copper, nickel, and cobalt are considered nearly pure metals. By continuously generating micro-plasma bubbles while scanning the device, arbitrarily shaped patterns, such as I and S shapes, could be written on the glass and epoxy resin substrate. Figure 8 shows typical writing patterns. Thus, we confirmed the feasibility of the proposed method as an on-demand metal deposition method.

## 5. Discussion

The focus of our research is to develop a new 3D wiring method to replace conventional methods, such as electro/electroless plating, PVD, CVD, and inkjet. To achieve this goal, we studied a metal deposition method using micro-plasma bubbles generated using an applied pulse. In this method, only a bubble injector and a metal ion solution were used to electrically write under atmospheric pressure and room temperature. Therefore, we can use the method for wiring on materials that cannot be plated by electrolytic plating, which requires conductive materials, or by PVD or CVD, which require reduced pressure. Moreover, because we can use patterning by scanning, a conventional patterning process is unnecessary. Thus, we can write on 3D surfaces independent of the substrate material properties, such as conductive/non-conductive and dry/wet. We previously proposed a metal deposition method using a suspension of metal nanoparticles [20]; however, as with the inkjet metal deposition method, the metal source was limited to the type of nanoparticles. However, in the process presented in this study, we can easily switch the metal source because metal ions are reduced in solution and deposited on a target. Furthermore, because this method uses an ionic solution as the metal source, it is expected to be applicable to the deposition of alloys and metal–polymer composite materials by adjusting the composition of the solution.

This study focuses on the generation mode of micro-plasma bubbles and metal deposition under pulse conditions because the generation mechanism of micro-plasma bubbles is still unclear. These micro-plasma bubbles are critically different from previously reported metal nanoparticle synthesis using plasma reactions in liquids [15,16] because of the deposition on a substrate.

In the metal deposition system developed in this study, the critical step is to generate on-demand micro-plasma bubbles by concentrating a high electric field at the tip of the bubble injector. We experimentally confirmed that the generation mode and metal deposition of micro-plasma bubbles can be controlled by adjusting the power supply conditions. As a result, we achieved metal deposition on a variety of dry/wet and conductive/nonconductive materials, such as chicken tissue and nitrile rubber. In addition, we were able to write copper patterning with an approximate 1 mm width on glass and epoxy resin substrates. The wiring resolution did not change significantly depending on the pulse conditions, potentially because the diameter of the deposits was not influenced by the bubble diameter, and was instead influenced by the electric field region, hence the discharge range. Considering that the deposition speed with this method was approximately 0.2–1 mm diameter patterning with an applied time of approximately 30 ms, the deposition is expected to be improved to approximately 3–16 mm/s when the device is moved 100–500 µm at a time in combination with robot scanning. This deposition time is very slow considering inkjet can write at approximately 1–4 m/s [27,28]. However, the metal deposition method we developed has the potential to be applied to non-conductive materials and wet materials, such as hydrogels and biological tissues, which are difficult to pattern directly using conventional patterning technologies. This will enable the integration of sensors and actuators into structures made of various materials, which is expected to advance soft robotics and biomimetics.

The most important result of this study is that micro-plasma bubbles in the APSB mode, where a single bubble is produced using all pulses, are suitable for metal deposition. This may be attributed to the intermittent generation of plasma, which increases the amount of reduction and deposition. Further understanding of the mechanism of metal deposition using micro-plasma bubbles is fundamental to practical metal deposition with higher resolution and the development of new functional devices by deposition of alloys and polymer–metal composites. For this purpose, it is necessary to study the solution composition and device design under high-speed observations of the deposition. In particular, we used the reduction effect of hydrated electrons and hydrogen atoms among the ionic and active species generated using micro-plasma bubbles to deposit metal. Still, we have not specifically approached the oxidation effect of hydroxyl radicals and oxonium ions. Therefore, investigating these oxidizing species is a possible way to solve the above problem. The conditions for generating micro-plasma bubbles in APSB mode need to be carefully studied. In this paper, higher voltage, longer pulse width, and a shorter off time are essential. However, these factors will increase the Joule heat and decrease the thermal diffusion time, which may make the application to hydrogels and biological samples difficult. The bubble generation mode may be determined by the timing of the next pulse in the bubble generation/expansion/collapsing process. Therefore, the problem can be solved by maintaining the APSB mode at lower voltages, shorter pulse widths, and longer off times by increasing the duration of these processes and controlling the solution properties. To incorporate sensors and actuators using this technology, it is necessary to fully evaluate the resistance of the deposited patterns and their adhesion to the substrate. Electron microscopy revealed that the surface of the deposited metal was not uniform, with cracks and unevenness. This is a fatal problem for the functionality of electric circuits. This problem may be solved by the approach to oxidized species described above. Furthermore, a combination of plasma and electroless plating is one of the possible solutions. Regarding adhesion plasma treatment of substrates, it can generally improve adhesion by adding functional groups to the surface and roughening the surface [29]. Therefore, this method may also have excellent adhesion properties. We consider that this method can be further developed by studying the above-mentioned solution composition, device design, and plasma modification.

## 6. Conclusions

We developed a new metal deposition method for various materials. We proposed a method for reducing metal ions in solution and depositing them on a target substrate using micro-plasma bubbles. We classified the generation mode of micro-plasma bubbles by applied pulse and confirmed that metal deposition was possible in APSB mode, in which one bubble was generated in all pulses. We also successfully used this method to deposit copper on dry/wet materials, such as chicken tissue and glass substrates. In addition, metal deposition of copper, nickel, chromium, cobalt, and zinc was achieved by simply changing the metal ion solution. Furthermore, copper patterns were drawn on glass and epoxy resin substrates. These results suggest that our method can be used as a 3D deposition method independent of the substrate material, such as conductive/non-conductive or dry/wet materials. In future work, we plan to measure the volume resistivity of the fabricated lines and investigate the approach to the oxidation reaction using micro-plasma bubbles, the solution, and device conditions for the APSB mode.

## Figures and Tables

**Figure 1 micromachines-13-01312-f001:**
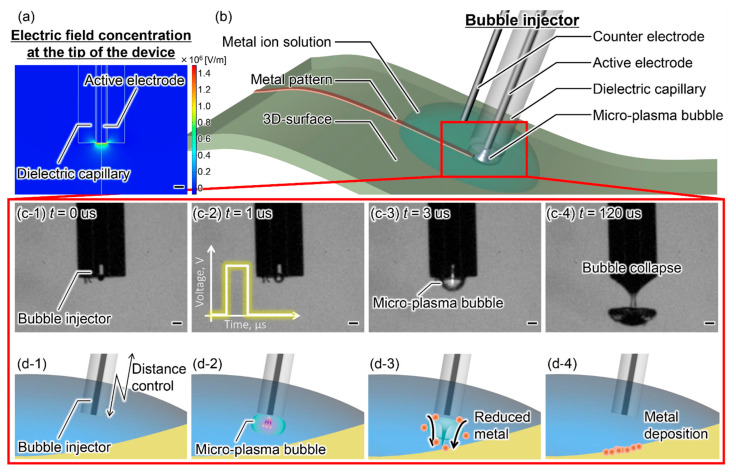
Overview of the metal deposition method using micro-plasma bubbles. All scale bars are 100 µm. (**a**) Finite element analysis of the electric field at the tip of the bubble injector. The active electrode is tungsten, the dielectric capillary is glass, and the voltage is 700 V. (**b**) Components of the proposed method and the concept of metal patterning. (**c**) Sequence of micro-plasma bubbles. The solution is a 0.9% sodium chloride solution, the voltage is 1000 V, and the pulse width is 10 µs. (**d**) Concept of the metal deposition process using micro-plasma bubbles.

**Figure 2 micromachines-13-01312-f002:**
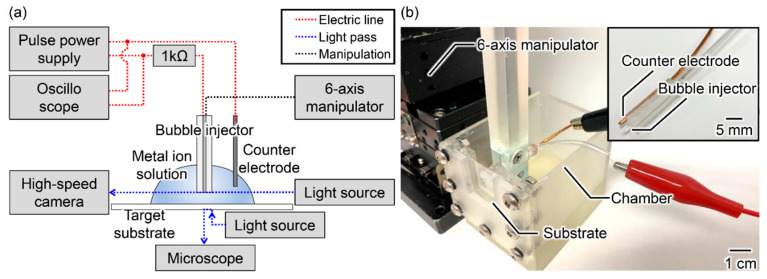
Overview of the constructed observation system for metal deposition using micro-plasma bubbles. (**a**) Schematic overview of the micro-plasma bubbles. (**b**) Photograph of the experimental apparatus.

**Figure 3 micromachines-13-01312-f003:**
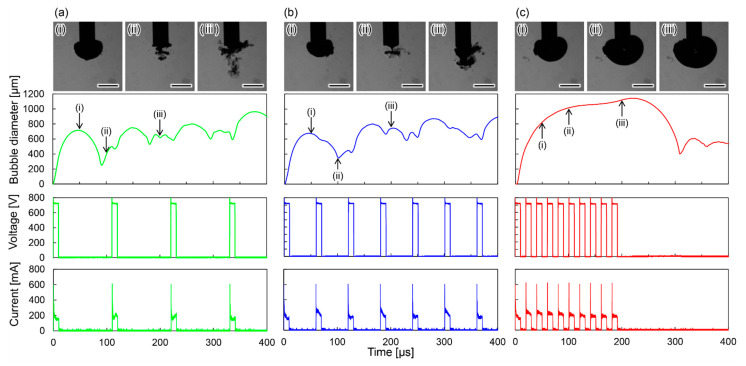
Photographs of micro-plasma bubbles at (**i**) 50 µs, (**ii**) 100 µs and (**iii**) 200 µs after application of a pulse and time series data of the bubble diameter, voltage and current. All scale bars are 100 µm. The voltage is 700 V, the pulse width is 10 µs, and the number of pulses is 10. In (**a**–**c**), the off times are 100 µs, 50 µs and 10 µs, respectively.

**Figure 4 micromachines-13-01312-f004:**
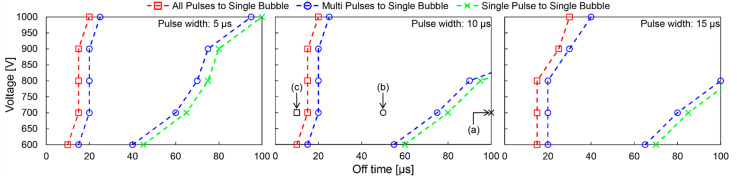
Results of the classification of micro-plasma bubble generation modes by pulse conditions. There are three generation modes: APSB, MPSB, and SPSB, and their boundaries are plotted and connected by dashed lines. (**a**–**c**) in the figure correspond to Figure 3.

**Figure 5 micromachines-13-01312-f005:**
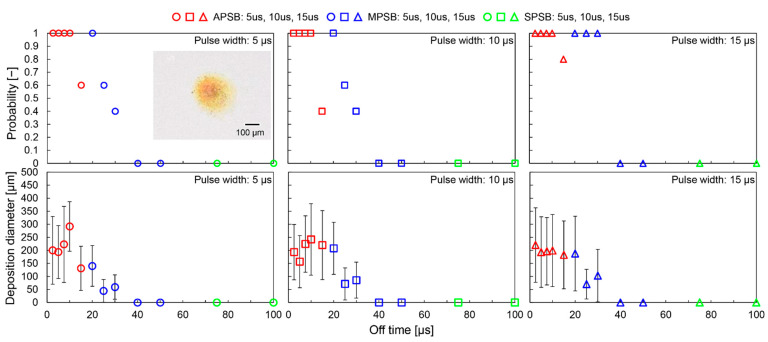
Results of the deposition probability and diameter with varying off time. The plots are color coded according to the bubble generation mode in Figure 4. The voltage was 700 V.

**Figure 6 micromachines-13-01312-f006:**
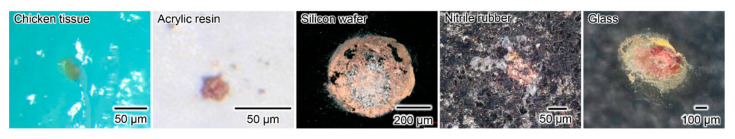
Copper deposition on various material surfaces (chicken tissue, acrylic resin, silicon wafer, nitrile rubber, glass). A copper sulfate solution was used as the solution, and Type A was used as the device.

**Figure 7 micromachines-13-01312-f007:**
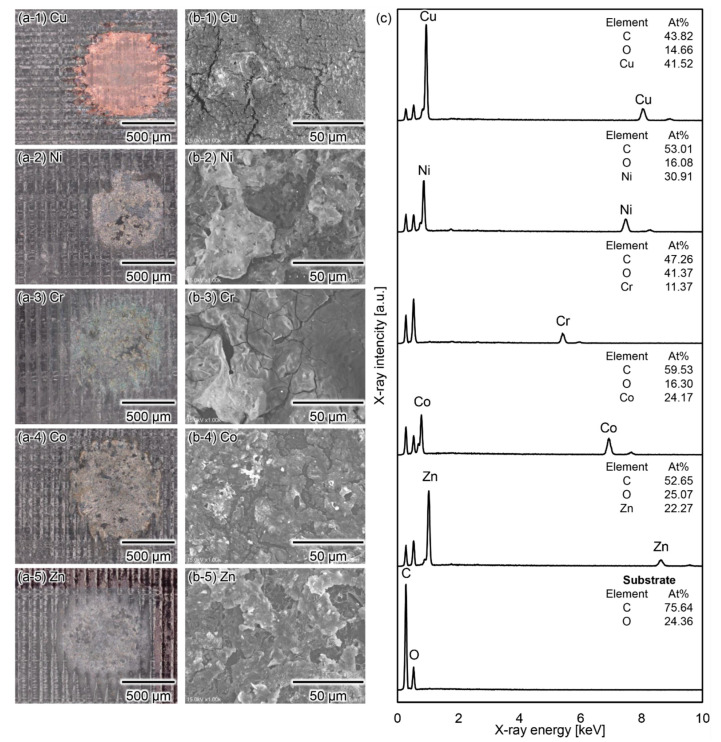
Deposition results of various metal species using micro-plasma bubbles. (**a**) Optical microscope image and (**b**) electron microscope image of deposition results (1-copper (Cu), 2-nickel (Ni), 3-chromium (Cr), 4-cobalt (Co), 5-zinc (Zn)) on epoxy resin. The solutions are aqueous acetate solutions, and the devices are Type B. (**c**) Elemental analysis of the deposit in (**b**) and substrate using energy dispersive X-ray spectroscopy. In all experiments, the power supply conditions were set to APSB mode.

**Figure 8 micromachines-13-01312-f008:**
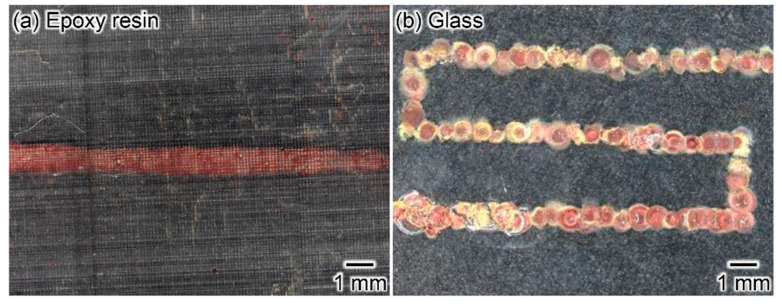
A demonstration of copper patterning. (**a**,**b**) Metal deposition by scanning the bubble injector in a I and S shape.

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
