# Peer review of "On-Demand Metallization System Using Micro-Plasma Bubbles"

_micromachines, 2022, doi:10.3390/mi13081312_

Round 1

Reviewer 1 Report

The title of this paper is On-demand metallization system using micro-plasma-bubbles. In this paper, the author propose a new metal deposition method using only a micro-plasma-bubble injector and a metal ion solution. Micro-plasma-bubbles were generated on demand using pulses, and the localized reaction field enables metal deposition independent of the substrate. Three different modes of micro-plasma-bubble generation were created depending on the power supply conditions and mode suitable for metal deposition. However, some problems need to be revised as follows:

1. Figure 1c shows the real-time status at different times. Only (c-1) has dimensions. It is recommended to add the following scales,

2. In section 3.3, the article wrote that the number of pulses should be the product of the pulse width and the number of pulses to be 10ms. Is there any problem in the interpretation of the number of pulses in this section?

3. In the description of Figure 3, the article writes that the pulse interval is 100µs, 50µs and 10µs respectively, and in SPSB mode, the bubble diameter reaches 700µm when the pulse is applied for 50µs in SPSB mode, and the bubble bursts before the next pulse, which means that it bursts at 60µs , which is not reflected in Figure 3a.

4. Since the article states that the pulse interval is 100µs, 50µs, and 10µs, the illustration of the phenomenon should reflect the described pulse interval.

5. For the description of Fig. 7, Fig. 7(c) shows whether the deposition can be indicated by lighting, but how to determine the learned element is indicated by the LED

6. There is a sentence problem in the language of the article, please carefully try to modify it

Reviewer 2 Report

Reviewer comment: Thanks for inviting me to review this paper titled "On-demand metallization system using micro-plasma-bubbles". In this paper, the authors used a micro-plasma-bubble injector and a metal ion solution to propose a new metal deposition. This work looks interesting and will be helpful for the possible reader of the "Micromachines". Therefore, I recommend publishing this research paper in the "Micromachines", but only after a manuscript revision. Please find my comments below.

1.   In the introduction section, please explain the advantages of the micro-plasma-bubbles system over other existing plasma systems, notably for this current work.

2.   The voltage and current waveforms should be added to understand discharge formation.

3.   As the authors used a glass substrate, it would be better if authors could provide the hydrophobicity results before and after the surface modification using ID water droplet onto the substrate and explain why hydrophobicity changes after surface modification.

4.   The surface morphology of the coated samples is unclear; therefore, it is recommended to study surface morphology using FE-SEM. EDS can also be added as a support for metal deposition.

5.   In this paper, the authors mainly discuss the phenomenon rather than intense discussion. If possible, please explain more about the reaction mechanism, especially how micro-plasma-bubbles help to form the metal nanoparticles.

      ……………….Good luck…………………

Round 2

Reviewer 1 Report

The authors' response is clear and the manuscript is suited for accretion.

Reviewer 2 Report

Decision:

The authors have revised the manuscript accordingly. Now the manuscript is ready to be published.